# Supplementary Motor Area Syndrome after Removal of an Unusual Extensive Parasagittal Meningioma: Analysis of Twelve Reported Cases

**DOI:** 10.3390/medicina58081126

**Published:** 2022-08-19

**Authors:** Chia-Chih Tsai, Yu-Feng Su, Feng-Ji Tsai, Hui-Yuan Su, Huey-Jiun Ko, Yung-Han Cheng, Yu-Li Chen, Cheng-Yu Tsai

**Affiliations:** 1Division of Neurosurgery, Department of Surgery, Kaohsiung Medical University Hospital, Kaohsiung 80708, Taiwan; 2Post Baccalaureate Medicine, College of Medicine, Kaohsiung Medical University, Kaohsiung 80708, Taiwan; 3Graduate Institute of Clinical Medicine, College of Medicine, Kaohsiung Medical University, Kaohsiung 80708, Taiwan; 4Division of Trauma and Surgical Critical Care, Department of Surgery, Kaohsiung Medical University Hospital, Kaohsiung 80708, Taiwan; 5Department of Public Health, Kaohsiung Medical University, Kaohsiung 80708, Taiwan; 6Graduate Institute of Medicine, College of Medicine, Kaohsiung Medical University, Kaohsiung 80708, Taiwan; 7School of Medicine, College of Medicine, Kaohsiung Medical University, Kaohsiung 80708, Taiwan

**Keywords:** supplementary motor area syndrome, extra-axial tumor, parasagittal meningioma, mutism aphasia, epilepsy

## Abstract

*Background and Objectives*: Supplementary motor area (SMA) syndrome is a common post-operation complication in intra-axial brain tumors, such as glioma. Direct damage to parenchyma or scarification of the major vessels during an operation are the main causes. However, it is rarely reported as a postoperative complication in extra-axial tumors. *Materials and Methods*: We reviewed 11 reported cases of supplementary motor area syndrome after removal of extra-axial meningiomas in the English literature from the PubMed database. We also added our case, which presented as an unusual huge meningioma, to analyze the clinical parameters and outcomes of these 12 reported cases. *Results*: Recovery time of supplementary motor area syndrome in extra-axial tumors could be within 1–7 weeks, shorter than intra-axial tumors (2–9 weeks). Epilepsy and progressive limb weakness are the most common presentations in 50% of cases. Different degrees of postoperative muscle power deterioration were noted in the first 48 h (from 0–4). Lower limbs (66.6%, 8/12) were slightly predominant compared to upper limbs (58.3%, 7/12). Mutism aphasia was also observed in 41.6% (5/12, including our case), and occurred in tumors which were involved in the dominant side; this recovered faster than limb weakness. *Discussion and Conclusions*: Our work indicated that SMA syndrome could occur in extra-axial brain tumors presenting as mutism aphasia and limb weakness without any direct brain parenchyma damage. In our analysis, we found that recovery time of postoperative motor function deficit could be within 1–7 weeks. Our study also provides a further insight of SMA syndrome in extra-axial brain tumors.

## 1. Introduction

Parasagittal meningioma is viewed as an extra-axial brain tumor, defined by the filling of the parasagittal angle, and usually involving the superior sagittal sinus (SSS). Clinical symptoms/signs present as seizure, headache, hemiparesis, personality change and behavior disturbance, depending on the relationship between tumor location and involved function area. Surgical resection is the standard treatment. Common post-operation complications are various, including wound breakdown/infection, cerebrospinal fluid leak/hydrocephalus, hematoma, venous infarct/cerebral edema and air embolism [1,2]. However, supplementary motor area syndrome (SMA syndrome) is rarely reported as a post-operation complication of parasagittal meningioma in the English literature.

SMA syndrome is widely reported in the resection of intra-axial tumors in SMA parenchyma, such as low-grade glioma [3]. SMA syndrome presents as rapid deterioration of the motor function after operation. Direct damage or destruction of the SMA parenchyma could be the cause, and a longer recovery time (2–9 weeks) has been reported [4,5,6,7,8,9]. However, SMA syndrome is rarely reported as a post-operation complication in extra-axial tumors due to no direct SMA parenchyma damage. To best of our knowledge, only 11 cases have been reported in the English literature [4,5,6,7,8].

Herein, we reported a case of a 55-year-old, right-handed male patient with an unusual huge parasagittal meningioma. Rapidly developing post-operation SMA syndrome was noted. Furthermore, we analyse and discuss these 12 cases, including our case, to comprise a small series of post-operation SMA syndrome after removal of extra-axial brain tumors.

## 2. Materials and Methods

### 2.1. Our Case Description

Our case was a 55-year-old, right-handed male patient with tonic-clonic jerk and consciousness loss as initial clinical presentations. Neurological examination showed gross normal finding, except for muscle power of 4+ over bilateral upper limbs. Brain computed tomography (CT) with reconstruction image was performed (Figure 1A,B). Then, brain magnetic resonance imaging (MRI) revealed a huge extra-axial tumor involved scalp and bilateral frontal bone with encasement of the superior sagittal sinus (SSS) (Figure 2A,B). Parasagittal meningioma was initially diagnosed under complete image study. Pre-operative angiography also showed no evidence of blood flow in the SSS within the lesion, meaning total occlusion of the SSS. Moreover, collateral vessels were also noted. Bilateral extended craniectomy and tumor removal were performed. During operation, an infiltrative tumor mass, which penetrated out of the skull bone and invaded into the underlying scalp tissue layer, was found. The tumor was easily bleeding and it contained many relative firm and elastic fibrous bands. Piecemeal removal of the tumor was performed. Although the tumor was severely adherent to the brain parenchyma, the pial surface was grossly intact on both sides of the hemisphere at the end of surgery. Moreover, major collateral veins were well preserved. The SSS was ligated and removed due to total occlusion of the SSS. After the tumor was totally removed, involved dura was resected and artificial material dura was water-tight sutured back in order to seal off the dura layer. A drainage tube was inserted in both the epidural and the subgaleal layer. The total operation time was 6 h. Simpson grade II was performed. While the patient returned to clear consciousness, extubation was performed in the intensive care unit after operation.

However, muscle powers of bilateral lower limbs progressed to downhill from grade 4 to 0, and the muscle power of the left upper limb was also progressed to 2 within 24 h after operation. Muscle tone was still preserved. Emergency brain CT demonstrated no hemorrhage, but mild edematous change of right vertex frontal-parietal lobe (Figure 3). Then, brain MRI was performed due to several episodes of intermittent seizure and poor recovery of muscle power post-operation. Brain MRI with T1 contrast revealed gross total removal of the meningioma without hematoma or obvious residual tumor to compress the surface of the brain parenchyma (Figure 4A,B). In addition, brain MRI with diffusion-weighted image (DWI) and apparent diffusion coefficient (ADC) revealed no obvious cortical damage in the SMA territory, nerve tract injury or acute infarction were found (Figure 5A,B). Mutism was also noted 2 days after the operation. Due to clinical presentations and image findings, postoperative SMA syndrome was diagnosed. Steroids and mannitol were used for mild brain edematous change. Prophylactic antibiotics were also used. Unfortunately, severe septic shock developed due to incidental diagnosis of diabetes mellitus without sufficient control before the operation. Third-line antibiotics and inotropic agents were used. Vital signs and general condition returned to stable. However, incidental epidural hematoma (EDH) occurred after the removal of the drainage tube on the tenth day post-operation. Due to the progressive poor consciousness level, emergency EDH removal was performed, and the patient’s consciousness recovered after EDH removal. However, weakness in four limbs and mutism aphasia were still noted.

After EDH removal, general condition and consciousness level returned to a stable status, but the muscle power of limbs and mutism aphasia still showed no improvement in the intensive care unit. A rehabilitation program was recommended. Two weeks after the tumor removal and rehabilitation, mutism aphasia was resolved to normal level. Seven weeks after tumor removal and rehabilitation, muscle power of the left upper limb improved from grade 1 to 5, and muscle power of both lower limbs returned from grade 1 to 4. Final pathology report illustrated meningioma, WHO grade I, which involved not only the meninx, but also the scalp and skull.

### 2.2. Literature Review and Case Analysis

We used SMA syndrome and falx/parasagittal meningioma as keywords for an English literature search in PubMed. Including our case, only 12 cases of the development of SMA syndrome after falx/parasagittal meningioma resection have been reported in the literature (Table 1). Inclusion criteria were SMA syndrome and parasagittal meningioma. Exclusion criteria were children and non-English literature.

## 3. Results

### Literature Review and Case Analysis

In analysis of these 12 reported cases, initial presentations of SMA were diverse, and dependent on the location, size and their relationship with the SSS. Epilepsy and progressive limb weakness were the most common presentations in 50% of cases (6/12, including our case). All 12 cases developed SMA syndrome immediately after operation. Different levels of postoperative muscle power deterioration was noted in the first 48 h, from 0–4. Lower limbs (66.6%, 8/12) were slightly more involved than upper limbs (58.3%, 7/12). Mutism aphasia was observed in 41.6% of cases (5/12, including our case), with tumors involving the left dominant hemisphere or the bilateral hemisphere. Muscle tone was preserved in all cases. Either brain CT and MRI were arranged after the surgery to rule out venous infarction or hemorrhage events on the brain. Mutism and muscle power could resolve to a normal level from several days to 7 weeks (our cases), and mutism aphasia often took priority. The recovery time of SMA syndrome in extra-axial tumors could be within 1–7 weeks.

## 4. Discussion

The supplementary motor area (SMA), namely the frontomesial area, localized in the posterior part of the superior frontal gyrus (Brodmann area 6), contributes to voluntary motor actions that demands the retrieval of motor memory and is critical to temporal sequences of actions. The SMA links not only to the primary motor cortex, but also the premotor cortex and cingulate cortex connect with the superior parietal lobe, insula, basal ganglia, thalamus, cerebellum and especially with the contralateral SMA through the corpus callosum [10]. It is due to the complex connection to both cortical and subcortical structures that means that damage to the area usually results in neurological deficit. An operation which involves the SMA may be followed by transient contralateral weakness and mutism aphasia. This complication was termed as SMA syndrome, and initially was difficult to distinguish from sequelae from injury to the primary motor cortex. Compared to damage to the primary motor cortex, SMA syndrome usually preserves muscle tone and can recover in days to weeks. SMA syndrome was widely reported in intra-axial tumor resection, such as glioma. The rate was approximately 11–100%. However, SMA syndrome was rare reported in extra-axial tumor removal in the English literature due to not being directly involved in the brain parenchyma.

The mechanism of developing SMA syndrome after the intervention of an extra-axial tumor is still unclear. Several hypotheses have been proposed, and include local oedema, the use of retractors, microvascular damage, and resection of healthy parenchyma [8]. According to our results, microvascular damage could be the most possible reason. Several articles suggested maintaining dissection within the arachnoid plain, operating with a hanging head, and using a navigation system during the surgery to prevent SMA syndrome [11]. Multidisciplinary treatment strategies are mandatory for post-operative treatment. Medical treatment, such as steroid and osmotic agents, were recommended to resolve brain edematous change. Craniectomy is also an essential option for refractory intracranial hypertension. Moreover, a rehabilitation program of directly stimulating the contralateral SMA or indirectly stimulating the adjacent cortex of affected SMA is also an important step for the recovery therapy.

Regarding our case and the cases of our analysis, 11 cases were parasagittal meningiomas and 1 case was falx meningioma (Table 1). The mean age was 59 y/o (range from 48–77 y/o), and the male/female ratio was 1:1. The mean tumor size was 59.8 mm (range from 31 mm–91 mm). According to the previous reports, SMA syndrome recovered in 2–9 weeks after intra-axial tumor excision [12]. In our study, motor function recovered within 1–7 weeks. The recovery period was shorter than intra-axial tumors due to no direct involvement or damage of the SMA parenchyma. In addition, mutism aphasia was normally resolved faster than muscle strength, and it only presented in tumors involving the dominant hemisphere. Indeed, 41.6% (5/12) of patients had a tumor on the dominant hemisphere, and all of them developed mutism immediately after the surgery. Collectively, the concept for the recovery pattern of aphasia mutism and motor function is reasonable, and can result in sufficient recovery after intensive rehabilitation program training.

## 5. Conclusions

Our finding indicated that supplementary motor area syndrome could occur in an extra-axial brain tumor, presenting as mutism aphasia and motor deficit without any direct brain parenchyma damage or collateral circulation. In addition, recovery time of postoperative motor function deficit could be within 1–7 weeks. Our study also provides a further understanding of SMA syndrome with extra-axial brain tumors.

## Figures and Tables

**Figure 1 medicina-58-01126-f001:**
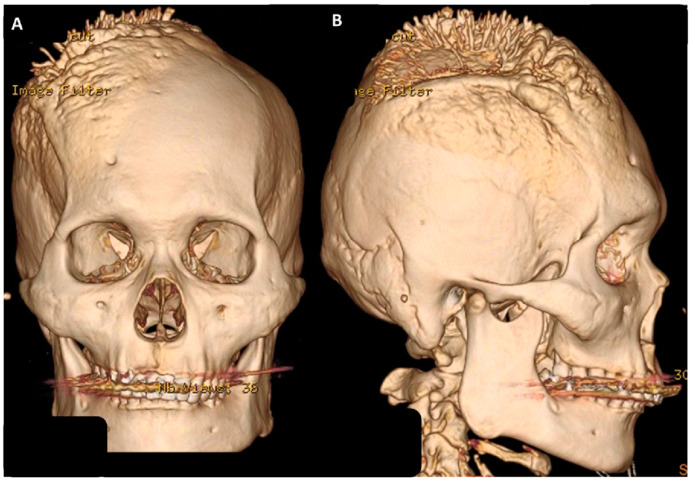
Brain CT with reconstruction image showed a huge extra-cranial osteophyte lesion over right skull bone: (**A**) Anterioposterior (AP) view; (**B**) lateral view.

**Figure 2 medicina-58-01126-f002:**
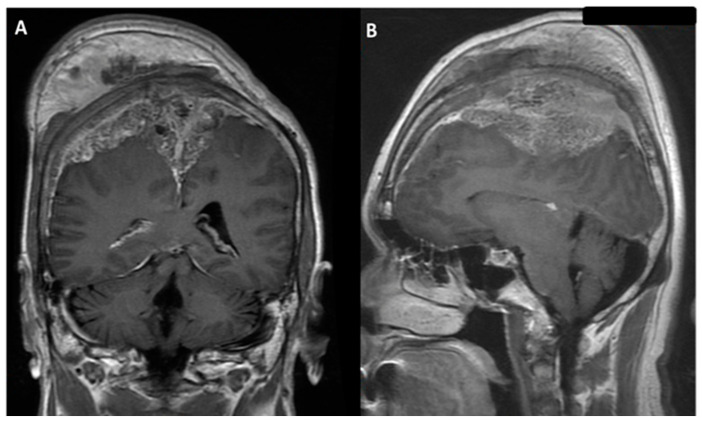
Brain magnetic resonance imaging (MRI) with gadolinium image revealed enhancement of an extra-axial tumor, involving scalp and bilateral frontal bone with encasement of the superior sagittal sinus; parasagittal meningioma was impressed: (**A**) coronal view; (**B**) sagittal view.

**Figure 3 medicina-58-01126-f003:**
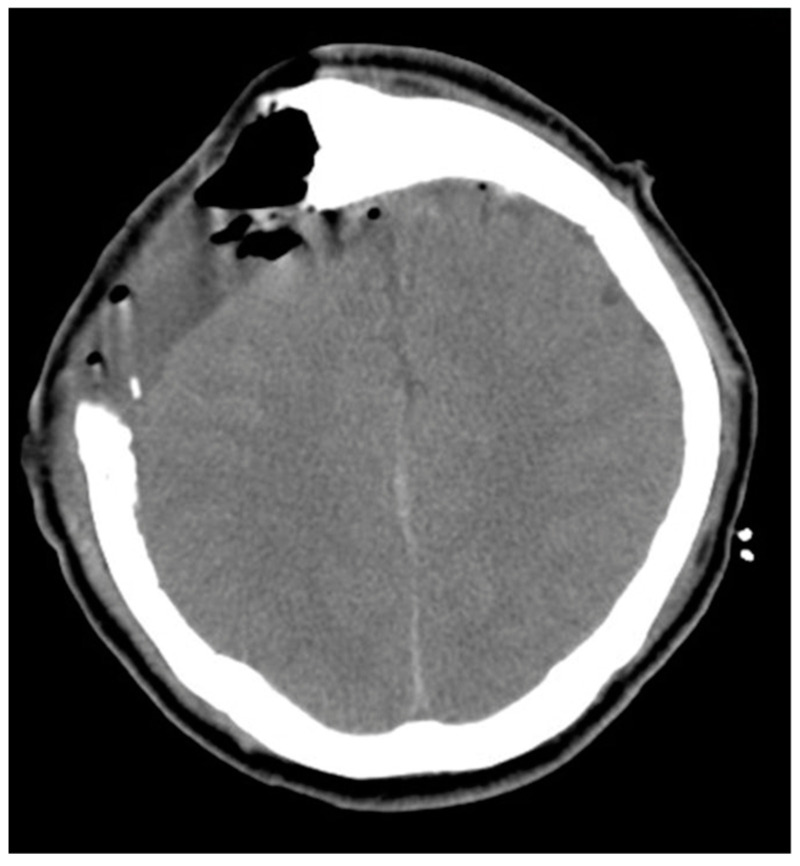
Post-operation brain CT image revealed no hemorrhage, but mild edematous change of the right vertex frontal-parietal lobe.

**Figure 4 medicina-58-01126-f004:**
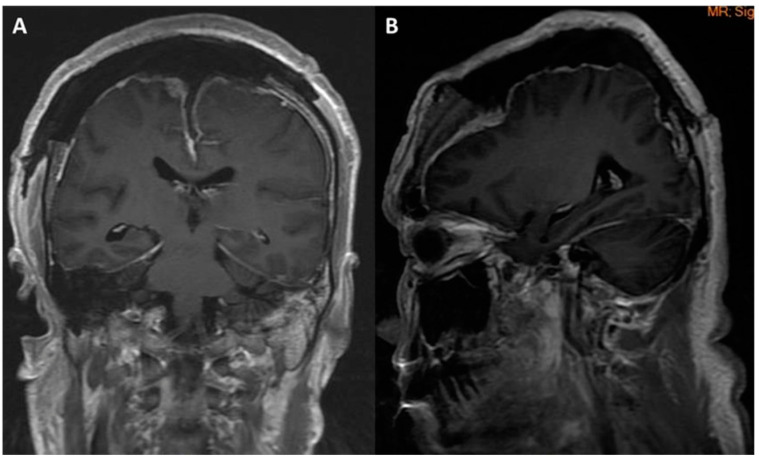
Post-operation brain MRI image revealed gross total tumor removal without hemorrhage or infarction: (**A**) coronal view; (**B**) sagittal view.

**Figure 5 medicina-58-01126-f005:**
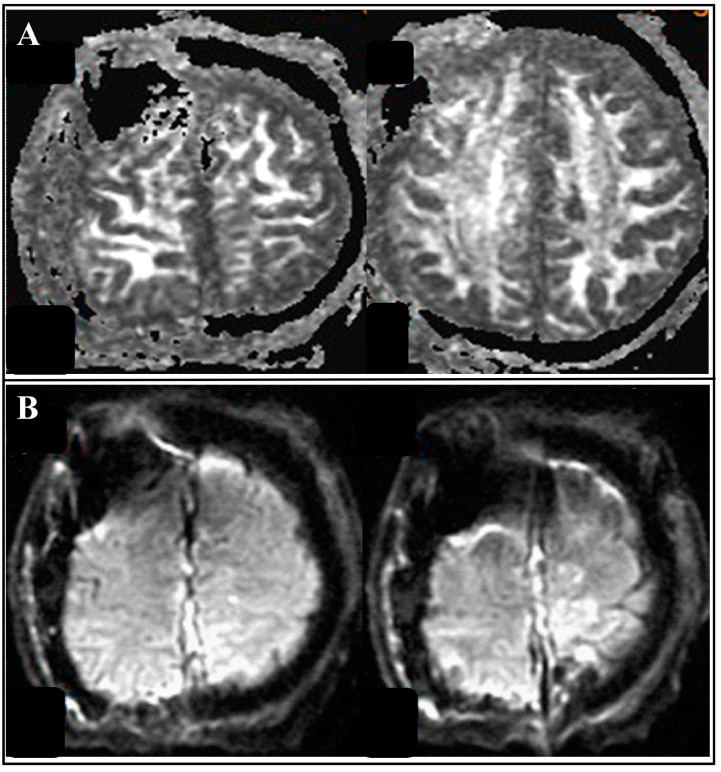
Follow-up brain MRI with diffusion tensor imaging(DTI). (**A**) brain MRI with diffusion-weighted image (DWI) revealed no tract injury or parenchyma damage. (**B**) brain MRI with apparent diffusion coefficient (ADC) revealed no acute infarction.

**Table 1 medicina-58-01126-t001:** Twelve cases of SMA syndrome in extra-axial tumors.

Reference	Age/Sex	Preoperative Symptoms	Tumor Size (mm)	Tumor Location	Sagittal Sinus	Simpson Grade	Pathology Grade	Postoperative	Muscle Tone	Speech	Follow-Up
								Motor Function			
Heiferman et al., 2014 [4]	42/F	Headache,	70	bilateral	occluded	II	II	4/5 upper limbs	N/A	Global aphasia	Start to move after 16 days
		mild weakness						2/5 lower limbs			
	53/F	Seizure	65	bilateral	occluded	II	I	0/5 upper limbs	N/A	Aphasia	Start to move 15 days
								0/5 lower limbs			
Satter et al., 2017 [5]	N/A	N/A	N/A	right	N/A	N/A	N/A	N/A	N/A	N/A	Relatively slow process
Nigro and Delfini, 2018 [6]	N/A	N/A	N/A	left	N/A	N/A	N/A	N/A	N/A	Dysphasia	N/A
Berg-Johnsen and Høgestøl, 2018 [7]	49/F	Headache, dizziness, reduced memory	40	bilateral	occluded	I	II	2/5 left foot and ankle	Preserved	Normal	Recovery after 3–6 months
	49/M	Epileptic seizures, palpable bump	50	left	occluded	II	II	2/5 right leg	Preserved	Speech arrest	Aphasia recovery after a few days
	70/M	Epileptic seizure, mental changes	68	right	ingrowth	II	I	3/5 left arm and leg	Preserved	Normal	Complete recovery of function
	77/M	Epileptic seizure, mental changes	64	left	open	I	II	3/5 right arm and leg	Preserved	Speech arrest	Aphasia recovery after a few days
	68/M	Epileptic seizure, mental changes	31	right	open	II	I	4/5 left arm and leg	Preserved	Normal	Complete recovery of function
Martinez-Perez et al., 2020 [8]	48/F	N/A	N/A	left	N/A	II	N/A	2/5 right arm	N/A	Global aphasia	Aphasia recovery after 6 days; muscle power recovery after 15 days
	49/F	N/A	n/A	right	N/A	II	NA	1/5 left body	N/A	N/A	Within 10 days recovery
Tsai et al., 2021	55/M	Seizure, loss of consciousness	91	bilateral	occluded	II	I	2/5 left arm and left leg 0/5 right leg	Preserved	Aphasia	Aphasia recovery within 7 days; muscle power recovery within 7 weeks

F, female; M, male; N/A, not available.

## Data Availability

Not application.

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
