# Peer review of "Supplementary Motor Area Syndrome after Removal of an Unusual Extensive Parasagittal Meningioma: Analysis of Twelve Reported Cases"

_medicina, 2022, doi:10.3390/medicina58081126_

Round 1

Reviewer 1 Report

Rereview of manuscript titled ”Supplementary Motor Area Syndrome after Removal of An Unusual Extensive Parasagittal Meningioma: Analysis of Twelve reported cases”. The authors have revised their manuscript according to previous comments, and I have no further concerns.

Author Response

Dear Reviewer

Thanks for your kindly reply 

Thanks you!!

Address correspondence: Dr. Cheng Yu Tsai

Division of Neurosurgery, Department of Surgery, Kaohsiung Medical University

Hospital, No.100 Tzyou 1st Road, Kaohsiung 80708, Taiwan

Telephone number: 886-7-3121101-6208

Fax number: 886-7-3127056

Reviewer 2 Report

Al my issue have been addressed, except  one reference I reccomended, guarracino et al. , on cognitive deficits in meningioma patients.

Author Response

Dear reviewer

Thanks for your kindly reply

We already added the reference that your mentioned in line 54

Address correspondence: Dr. Cheng Yu Tsai

Division of Neurosurgery, Department of Surgery, Kaohsiung Medical University

Hospital, No.100 Tzyou 1st Road, Kaohsiung 80708, Taiwan

Telephone number: 886-7-3121101-6208

Fax number: 886-7-3127056

This manuscript is a resubmission of an earlier submission. The following is a list of the peer review reports and author responses from that submission.

Round 1

Reviewer 1 Report

Thank you for the possibility to review the manuscript "Supplementary Motor Area Syndrome after Removal of An Un-2 usual Huge Parasagittal Meningioma: Analysis of Twelve re-3 ported cases" by Chia-Chih Tsai and co-authors.

Authors address SMA syndrome as a postoperative complication of extra-axial tumors, by reviewing 11 cases from the literature plus one single case studied by the authors themselves.

They report that recovery time is shorted that intra-axial surgery. The frequency of epilepsy and progressive limbs weakness is 50% (lower limbs 66.6%, vs upper limbs 58.3%), and mutism aphasia in 41.6%.

Authors argue that SMA syndrome could be found in extra-axial lesion and the study provides further insight in this issue.

I have the following observations to the authors' attention.

The introduction lacks of specific hypothesis.

In addition, there is one study that could be cited, presenting a series of meningiomas to the sensorymotor area (including SMA) in which neuropsychological deficits are reported (PMID: 31790726).

Methods

Authors report that their patient had aphasia. However no description about how this was diagnosed (i.e. neuropsychological tests?) is reported.

Please report on which literature search websites the literature search was done and the specific keywords used.

Please report the exclusion/inclusion criteria for the literature search

Please report how many papers were discarded and the reasons for dropping them out.

Discussion

Please report the potential mechanisms that could be responsible for the deficits reported in extra-axial lesions. Mechanical? Pressure? Or other hypothesis. (This issue is related to my point raised in the introduction about the lack of specific hypothesis).

Author Response

Reviewer 1.

Thank you for the possibility to review the manuscript "Supplementary Motor Area Syndrome after Removal of An Un-2 usual Huge Parasagittal Meningioma: Analysis of Twelve re-3 ported cases" by Chia-Chih Tsai and co-authors.

Authors address SMA syndrome as a postoperative complication of extra-axial tumors, by reviewing 11 cases from the literature plus one single case studied by the authors themselves.

They report that recovery time is shorted that intra-axial surgery. The frequency of epilepsy and progressive limbs weakness is 50% (lower limbs 66.6%, vs upper limbs 58.3%), and mutism aphasia in 41.6%.Authors argue that SMA syndrome could be found in extra-axial lesion and the study provides further insight in this issue. I have the following observations to the authors' attention.

Dear Reviewer:

Thank you for your valuable comments. We response your comments point by point as below:

Question1: The introduction lacks of specific hypothesis.

Answer 1: Your comment is well taken. We try to offer the possibility of SMA syndrome which was induced by extra-axial tumor. Moreover, we try to present the analytic result of this kind of syndrome under extra-axial tumor.

Question 2: In addition, there is one study that could be cited, presenting a series of meningiomas to the sensorymotor area (including SMA) in which neuropsychological deficits are reported (PMID: 31790726).

Answer 2: Thanks for your comment. We added 4-8 references in line 54 in introduction to collect and support our hypothesis

Methods

Question 3: Authors report that their patient had aphasia. However no description about how this was diagnosed (i.e. neuropsychological tests?) is reported.

Answer 3: Thanks for your comment. We just approach by clinical neurological examination

Question 4: Please report on which literature search websites the literature search was done and the specific keywords used.

Answer 4: Thanks for your comment. We used SMA syndrome and falx/parasagittal meningioma as keywords for English literatures search in PubMed in line 128-129.

Question 5: Please report the exclusion/inclusion criteria for the literature search

Answer 5: We added “Inclusion criteria were SMA syndrome and parasagittal meningioma. Exclusion criteria were children and non-English literatures in line 131-132.

Question 6: Please report how many papers were discarded and the reasons for dropping them out.

Answer 6: Thanks for your comment. No paper was dropped out due to so few literatures were reported

Question 7:

Discussion

Please report the potential mechanisms that could be responsible for the deficits reported in extra-axial lesions. Mechanical? Pressure? Or other hypothesis. (This issue is related to my point raised in the introduction about the lack of specific hypothesis).

Answer 7: Your comment is well taken. We added the sentence in discussion as

”According to our results, microvascular damage could be the most possible reason.” In line 182-183.

Address correspondence: Dr. Cheng Yu Tsai

Division of Neurosurgery, Department of Surgery, Kaohsiung Medical University

Hospital, No.100 Tzyou 1st Road, Kaohsiung 80708, Taiwan

Telephone number: 886-7-3121101-6208

Fax number: 886-7-3127056

Reviewer 2 Report

I have reviewed the paper entitled "Supplementary Motor Area Syndrome after Removal of An Unusual Huge Parasagittal Meningioma: Analysis of Twelve reported cases".

The title seems that the study reviewed self-managed 12 patients. But the reader notices that the 11 cases are from published papers but only one new is reported that decrease the originality of the paper.

I find nothing new regarding the findings and 11 of 12 patients are already published.

Best regards,

Author Response

Reviewer 2

I have reviewed the paper entitled "Supplementary Motor Area Syndrome after Removal of An Unusual Huge Parasagittal Meningioma: Analysis of Twelve reported cases".

The title seems that the study reviewed self-managed 12 patients. But the reader notices that the 11 cases are from published papers but only one new is reported that decrease the originality of the paper.

I find nothing new regarding the findings and 11 of 12 patients are already published.

Dear Reviewer:

Thanks for your comment. Although this is just one patient in 12 cases, we tried to analysis and offer new findings. Hope you could recognize our efforts. Thank you!!

Reviewer 3 Report

Review of the manuscript “Supplementary Motor Area Syndrome after Removal of An Unusual Huge Parasagittal Meningioma: Analysis of Twelve reported cases”. The authors describe a rare case of an extensive parasagittal meningioma and review the literature of SMA syndrome in extra-axial tumors.

The manuscript needs some overall formatting with regards to language and grammar.

Title

May I suggest using a synonym to “huge”, perhaps extensive?

Introduction

Please add reference to the recovery time after SMA in intra-axial tumors

Materials and methods

The sentence seems incomplete: “Parasagittal meningioma was initially diagnosed.”

This sentence also needs formatting: “tumor was easily bleeding and relative firm and elastic fibrous band content was noted.”

What did you use for dural plasty? Which Simpson grade would you grade the surgery at?

Did you remove the affected bone and perform a cranioplasty? How was the subcutaneous/subgaleal scalp mass handled?

The reason for the septic shock seems unclear to me, what was the suspected infectious focus?

Later a drain is mentioned, please add and the drain location (epidural, subgaleal) this under the description of the surgery.

Author Response

Reviewer 3:

Review of the manuscript “Supplementary Motor Area Syndrome after Removal of An Unusual Huge Parasagittal Meningioma: Analysis of Twelve reported cases”. The authors describe a rare case of an extensive parasagittal meningioma and review the literature of SMA syndrome in extra-axial tumors.

The manuscript needs some overall formatting with regards to language and grammar.

Dear Reviewer:

Thank you for your valuable comments. We response your comments point by point as below:

Question1: Title

May I suggest using a synonym to “huge”, perhaps extensive?

Answer 1: Thanks for your comment. We adjusted the title from huge to extensive in line 3.

Question2: Introduction

Please add reference to the recovery time after SMA in intra-axial tumors

Answer2: Thanks for your comment. We added reference [4-8] in line 54 in introduction

Question3: Materials and methods

The sentence seems incomplete: “Parasagittal meningioma was initially diagnosed.”

This sentence also needs formatting: “tumor was easily bleeding and relative firm and elastic fibrous band content was noted.”

Answer3: Thanks for your comment. We revised the sentence

“Parasagittal meningioma was initially diagnosed under complete image study.” And

“Tumor was easily bleeding and it contained many relative firm and elastic fibrous bands in line 71 and 76.”

Question4: What did you use for dural plasty? Which Simpson grade would you grade the surgery at?

Answer4: Thanks for your comment. We used DuraGen® Classic for dural plasty. We did Simpson grade II and we added it in manuscript in line 83.

Question5: Did you remove the affected bone and perform a cranioplasty? How was the subcutaneous/subgaleal scalp mass handled?

Answer5: Thanks for your comment. We removed all affected bone and craniectomy was done without cranioplasty. We also removed most of the subcutaneous/subgaleal scalp mass

Question6: The reason for the septic shock seems unclear to me, what was the suspected infectious focus?

Answer6: Thanks for your comment. I think immune-compromised (DM with poor control) maybe the main reason of septic shock.

Question7: Later a drain is mentioned, please add and the drain location (epidural, subgaleal) this under the description of the surgery.

Answer7: Thanks for your comment. Drainage was inserted both in epidural and subgaleal layer in line 82.

Address correspondence: Dr. Cheng Yu Tsai

Division of Neurosurgery, Department of Surgery, Kaohsiung Medical University Hospital, No.100 Tzyou 1st Road, Kaohsiung 80708, Taiwan

Telephone number: 886-7-3121101-6208

Fax number: 886-7-3127056

Round 2

Reviewer 2 Report

I have re-reviewed the paper entitled "Supplementary Motor Area Syndrome after Removal of An Unusual Extensive  Parasagittal Meningioma: Analysis of Twelve reported cases".

This paper is neither an original research article nor a literature review one. So even if it is considered for publication this might be regarded and re-edited as a case report. Because there is no new finding and it just add one more case to the field. The reader notices that the 11 cases are from published papers but only one new is reported that decrease the originality of the paper.

The paper might be accepted as a case report.

Author Response

Reviewer 2

I have re-reviewed the paper entitled "Supplementary Motor Area Syndrome after Removal of An Unusual Extensive Parasagittal Meningioma: Analysis of Twelve reported cases".

This paper is neither an original research article nor a literature review one. So even if it is considered for publication this might be regarded and re-edited as a case report. Because there is no new finding and it just add one more case to the field. The reader notices that the 11 cases are from published papers but only one new is reported that decrease the originality of the paper.

The paper might be accepted as a case report.

Dear Reviewer:

Thanks for your kindly reply and valuable comments. We realized the point that you concerned. However, we still think the work is not just only descriptive article. We performed critical literatures review and analysis for the result of several series, included our case. We also performed simple statistic method to analyze the results, such as recovery time and main clinical symptom/sign.

In addition, our work is still belonged to some concepts of origin article in this kind of expressive way, such as our previous article in other journal:

WORLD NEUROSURGERY 127: e311-e320, JULY 2019 Original Article

Meningeal Melanocytoma Associated with Nevus of Ota: Analysis of Twelve Reported Cases

Therefore, hope you could recognize our efforts. Thank you!!

Address correspondence: Dr. Cheng Yu Tsai

Division of Neurosurgery, Department of Surgery, Kaohsiung Medical University

Hospital, No.100 Tzyou 1st Road, Kaohsiung 80708, Taiwan

Telephone number: 886-7-3121101-6208

Fax number: 886-7-3127056
